# Enhancing Progestin Therapy with a Glucagon-Like Peptide 1 Agonist for the Conservative Management of Endometrial Cancer

**DOI:** 10.3390/cancers17040598

**Published:** 2025-02-10

**Authors:** Andrea R. Hagemann, Ian S. Hagemann, David G. Mutch, Eric J. Devor, Paige K. Malmrose, Yuping Zhang, Abigail M. Morrison, Kristina W. Thiel, Kimberly K. Leslie

**Affiliations:** 1Division of Gynecologic Oncology, Department of Obstetrics and Gynecology, Washington University, and Alvin J. Siteman Cancer Center, St. Louis, MO 63110, USA; hagemanna@wustl.edu (A.R.H.); hagemani@wustl.edu (I.S.H.); mutchd@wustl.edu (D.G.M.); 2Department of Pathology and Immunology, Washington University, St. Louis, MO 63110, USA; 3Department of Obstetrics and Gynecology and the Holden Comprehensive Cancer Center, University of Iowa, Iowa City, IA 52242, USA; eric-devor@uiowa.edu (E.J.D.); yuping-zhang@uiowa.edu (Y.Z.); abby-morrison@uiowa.edu (A.M.M.); 4Department of Pathology, University of Illinois at Chicago, Chicago, IL 60612, USA; pmalmr2@uic.edu; 5Department of Internal Medicine and the University of New Mexico Comprehensive Cancer Center, University of New Mexico Health Sciences Center, Albuquerque, NM 87131, USA

**Keywords:** endometrial cancer, hormone therapy, glucagon-like peptide 1 receptor agonists, conservative management, progesterone

## Abstract

Since obesity is a major risk factor for endometrial cancer, we explored a dual treatment strategy by combining a weight loss drug (semaglutide—a glucagon-like peptide-1 (GLP-1) agonist) with a progestin. Progestins are a safe and highly effective non-surgical treatment strategy for early-stage/grade endometrial cancer, but the response is often not sustained. Using cell lines and patient-derived preclinical models of endometrial cancer, we discovered an unexpected interplay between the GLP-1 receptor and progesterone receptor. Specifically, GLP-1 agonists increased expression of the progesterone receptor, and the combination of the two agents resulted in more pronounced cell death as compared to a single agent alone. As more patients are exploring GLP-1 agonists for weight management, these data make an important contribution to the field by suggesting that such drugs can also increase the efficacy of hormone therapy in endometrial cancer.

## 1. Introduction

Although obesity is a strong risk factor for many cancers, endometrial cancer (EC) is the malignancy most commonly associated with obesity [1]. Up to 90% of the 65,000 women diagnosed with endometrial cancer each year in the US are diagnosed as overweight or obese, and up to 60% of endometrial cancer cases are attributed to obesity [2]. Furthermore, predictive models show that anovulation, nulliparity, polycystic ovarian syndrome, insulin resistance, and diabetes—all correlates of obesity—increase the risk of atypical endometrial hyperplasia (AEH) in premenopausal women by six- to ten-fold [3,4]. Thus, as the obesity rate in the US has increased, so too has the rate of endometrial cancer [5].

For postmenopausal women or women not desiring fertility preservation, hysterectomy is highly effective in preventing endometrial cancer development in patients with atypical endometrial hyperplasia (AEH) and is frequently curable for women with early-stage/grade endometrial cancer (EC). However, for premenopausal women with AEH and/or early-stage/grade EC, removing the uterus would preclude pregnancy, whereas delaying hysterectomy could allow cancer to develop. Thus, such patients are often offered progestin therapy, usually via a levonorgestrel intrauterine device (LNG-IUD) or oral progestational agents. These patients often choose progestin therapy because of their desire for fertility. Although progestins are effective, the risk of relapse is as high as 50% within 6 months [6,7]. Thus, better strategies to prevent endometrial cancer in premenopausal women with AEH who desire pregnancy are needed.

In a normal cycling female, the endometrial lining of the uterus is dynamically regulated by a balance of estrogen and progesterone throughout the menstrual cycle. In the normal endometrium, estrogen provides a pro-proliferative signal, whereas progesterone counters the effects of estrogen to promote differentiation of the endometrial epithelial cells [4]. An estimated 80% of EC diagnoses are attributed to an imbalance in estrogen and progesterone—either too much estrogen or insufficient progesterone. Estrogen acting through estrogen receptors (ERs) α and β plays a key role in EC through transcriptional regulation of pro-growth signaling pathways, including recruitment of specific co-activators to estrogen-responsive elements. In addition, estrogen has non-genomic actions that contribute to oncogenesis, such as activation of the PI3K and MAPK pathways that are often aberrantly activated in EC [8]. In contrast, progesterone-mediated activation of the progesterone receptor (PR) serves as a tumor suppressive signal in the endometrium, including induction of differentiation, cell cycle arrest, and, potentially, apoptosis along with inhibition of inflammation and invasion [9,10,11,12,13,14,15,16,17]. Thus, targeting this hormone imbalance is a potentially safe strategy to prevent and/or treat EC.

Synthetic progesterone mimetics, termed progestins, have been explored for treatment of EC (reviewed in references [9,18,19]). Like progesterone, progestins are hypothesized to promote differentiation of tumor cells and potentially block cell division and/or induce apoptosis. There is a trend towards improved response to progestins with expression of PR. For example, the overall response rate in patients with expression of PR by immunohistochemistry (IHC) is 72%, whereas patients with no detectable expression of PR in the tumor epithelial cells have a response rate of 12% [20]. Unfortunately, a number of patients experience relapse after their initial response to progestin, indicating that further optimization of progestin therapy is necessary to attain a durable response in EC.

Progestins are synthesized from progesterone and androgen backbones. For example, levonorgestrel is a derivative of 17α-ethynyl-19-nortestosterone-18-methylestrane, and levonorgestrel binds to both PR and other steroid hormone receptors, including the androgen receptor (AR). Classic nuclear receptors are well-described transcription factors that, upon ligand binding, dimerize, translocate into the nucleus, and bind to hormone-responsive gene promoters. In addition to ligand binding, nuclear receptors can be activated by phosphorylation in a ligand-dependent or -independent manner. However, in addition to nuclear receptors, progesterone and progestins also activate more recently identified membrane receptors PGRMC1 and 2, which are activated in the cell membrane by ligand binding and enhance downstream signaling through molecules such as Src and Erk [21]. There is a potential synergy between Src and Erk signaling induced by PGRMC1 and 2, as these signaling molecules both phosphorylate PRB, which is the PR receptor isoform that resides in the cytoplasm in its inactive, unphosphorylated state. However, upon phosphorylation, PRB is bound by ligand and translates into the nucleus, thereby allowing PR gene transcription. Importantly, GLP-1R agonists also bind to membrane receptors that enhance Src and Erk signaling, and GLP-1R is reported to interact with and co-activate with PGRMC1, suggesting a synergistic effect on PR-mediated EC cell growth inhibition. Hence, in this study, we theorized that the combination of a GLP-1R agonist and a progestin would be useful as a novel treatment or prevention regimen and tested this hypothesis in preclinical models using EC cell lines and patient-derived organoids.

## 2. Methods

Drugs: Semaglutide, liragutide, and levonorgestrel were purchased from Selleck Chemicals and reconstituted in DMSO.

Cell lines: KLE endometrial cancer cells were purchased from the American Type Culture Collection (CRL-1622), while Ishikawa and Hec50 endometrial cancer cells were gifts from Dr. Erlio Gurpide (New York University). The origins and validation of these cell lines as models of endometrial cancer have been extensively reported [22,23]. Cell line identities were verified via CODIS DNA profiling at BioSynthesis, Inc., Lewisville, TX, USA. Ishikawa and Hec50co cells were cultured in DMEM supplemented with 10% fetal bovine serum (FBS) and 1% antibiotic [penicillin (p)/streptomycin (s)] (all from Thermo Fisher Scientific, Inc., Waltham, MA, USA). KLE cells were cultured in RPMI 1640 (Thermo Fisher Scientific, Inc.) supplemented with 10% FBS and 1% p/s.

Western blotting: Expression of GLP-1R was assessed in 20 µg of protein collected from whole cell lysates of Hec50, KLE, and Ishikawa cells by Western blotting. Protein concentrations were determined using the BCA Protein Assay Kit ( Thermo Fisher Scientific, Inc.) per the manufacturer’s instructions. Proteins were separated by SDS-PAGE with 10% acrylamide followed by transfer to nitrocellulose membrane. After blocking in 5% milk in Tris-buffered saline buffer (TBST), membranes were exposed to primary antibodies against GLP-1R (# bs-1559R, Bioss Antibodies, Woburn, MA, USA, 1:1000 dilution in 5% BSA in TBST, incubated overnight at 4 °C) or β-actin (#A1978, Sigma-Aldrich, St. Louis, MO, USA, 1:5000 dilution in 5% milk in TBST, incubated 1 hr. at room temperature), followed by exposure to either anti-rabbit-HRP for GLP-1R or anti-mouse HRP for β-actin (each secondary antibody was at a 1:10,000 dilution in 5% BSA in TBST for 1 h). Bands were visualized using HRP-conjugated chemiluminescence reagent (SuperSignal West Pico PLUS, Thermo Fisher Scientific, Inc.). Expression of GLP-1R in each cell line by densitometry (NIH Image J version 1.54m) was calculated relative to Ishikawa cells after normalization to β-actin loading control.

RNA extraction and quantitative polymerase chain reaction: Total cellular RNA was isolated using the RNeasy Plus Kit (QIAGEN, Germantown, MD, USA), and 500 ng of RNA was subjected to reverse transcription using SuperScript III Reverse Transcriptase and an oligo (dT) primer (Thermo Fisher Scientific, Inc.) to yield cDNA, followed by amplification with Power SYBR Green (Thermo Fisher Scientific, Inc.) using the Applied Biosystems Model 7900HT in the Genomics Division of the Iowa Institute of Human Genetics (IIHG). The following genes were assessed by quantitative PCR (qPCR): GLP1R, PGRMC1, PGRMC2, FOXO1, ER, AR, GR, MR, and PR, with 18S rRNA used as the housekeeping gene. The PrimerQuest primer design tool (IDT, Coralville, IA, USA) was used to design primer sequences. The sequences of forward and reverse primers are as follows:
**Gene**
**5′-SEQUENCE-3′****Tm**GLP1RForCAGGCTCGTTCGTGAATGT55.4
RevGCGATAACCAGAGCAGAGAAG55.3PGRMC1ForAGCAGGAGACTCTGAGTGACTG58.0
RevCCTCATCTGCGTACACAGTGGG56.7PGRMC2ForCGTGACCAAAGGCAGCAAGTTC59.1
RevCTCTAAGTGCATCTTTATCTAGGC52.8FOXO1ForCTACGAGTGGATGGTCAAGAGC57.1
RevCCAGTTCCTTCATTCTGCACACG58.4ERαForTGGGCTTACTGACCAACCTG57.1
RevCCTGATCATGGAGGGTCAAA54.4ARForATGGTCCCTGGCAGTCTCCAAA60.7
RevATGGTGAGCAGAGTGCCCTATC58.8GRForGGAATAGGTGCCAAGGATCTGG57.4
RevGCTTACATCTGGTCTCATGCTGG57.7MRForAAATCACACGGCGACCTGTCGT61.4
RevATGGCATCCTGAAGCCTCATCC59.4PGRForATCCTACAAACACGTCAGTGGGCA60.5
RevACTGGGTTTGACTTCGTAGCCCTT60.318S rRNAForAACCTTTCGATGGTAGTCGCCG59.4
RevCCTTGGATGTGGTAGCCGTTT57.6


Data were calculated as the fold change in expression relative to untreated samples at the same time point using the ΔΔCt method.

Immunohistochemistry (IHC): ER and PR expression in primary tissues from which the patient-derived organoid models were generated was assessed by IHC by the Comparative Pathology Laboratory at the University of Iowa (Iowa City, IA, USA). IHC was performed on sequential sections using the following primary antibodies from Dako (Agilent, Santa Clara, CA, USA): PR (PgR636); ER (M7047). Secondary antibody was goat-anti-mouse for both PR and ER. Stained sections were incubated in DAB chromogen substrate and counterstained with hematoxylin prior to mounting with coverslips. Two independent reviewers applied the modified H-score (range of 0–300) to score staining. Specifically, the modified H-score is determined by multiplying the intensity of tumor staining (0–3) by the percent of cells with staining (0–100%). Positive control tissue was scored as a 3+ for intensity and a 100% for percentage. To ensure tissue integrity, the presence of signal was assessed in the stroma and myometrium. Modified H-scores from each reviewer (N = 2 reviewers) were averaged for each slide to yield the average modified H-score. Representative images were acquired with an EVOS inverted bright field microscope at 20× magnification.

Patient-derived organoids: Patient-derived organoids (PDOs) were isolated in a study approved by the University of Iowa’s IRB, protocol #201809807, from patients with grade 1 endometrial cancer undergoing surgical resection and established based upon our previously reported methods [17,24,25]. Organoids with a diameter of >50 µm typically developed within 3–14 days after tumor processing. To assess drug sensitivity, organoid harvesting solution (Cultrex, R&D Systems, Minneapolis, MN, USA) was used to collect organoids. After digestion into single-cell suspensions using TrypLE Express (Gibco, Waltham, MA, USA), cells were resuspended in AdDE+++ medium supplemented with 10% Matrigel. Next, 10,000 cells were seeded into each well of a 96-well ultra-low attachment white plate (U-bottom). Organoids were treated with 100 nM of progesterone (P4), 100 nM of levonorgestrel (L), 100 nM of semaglutide (S), or L+S for 72 h in organoid culture media (29, 30) containing 1 nM of estradiol as an additive to the media. Viability was assessed using CellTiter-Glo 3D reagent (Promega, Madison, WI, USA) per the manufacturer’s protocol; luminescence was determined with a Gen5 Microplate Reader (BioTek, Agilent, Winooski, VT, USA). All the tests were conducted in triplicate. Data were normalized to control, which was set at 100% viability.

Statistical analyses: All data were analyzed with GraphPad Prism version 10. To assess statistical significance, we used either Student’s *t*-test or one-way ANOVA with Tukey’s multiple comparison test.

## 3. Results

### 3.1. GLP-1R Expression in Endometrial Cancer Cell Models 

As previously reported, GLP-1Rs are expressed in the endometrium, both in malignant and non-malignant tissues. Kanda et al. reported expression of GLP-1R in Ishikawa endometrial cancer cells, as well as in over 100 tissue samples from endometrial tumors [26]. We confirmed that Hec50, KLE, and Ishikawa endometrial cancer cell models also express GLP-1Rs by Western blotting (Figure 1A, Appendix A). Of interest, expression of GLP-1R was consistent over multiple cell lines representing endometrioid (Ishikawa, KLE) and serous (Hec50) models [22,23].

Since our goal is to assess the combination of a GLP-1R agonist with a progestin, in subsequent experiments, we focused on Ishikawa cells, which are endometrial cancer cells with PR expression [22,23]. GLP-1R transcription is significantly induced upon treatment of Ishikawa endometrial cancer cells with GLP-1R agonists liraglutide or semaglutide beginning at 24 h, with continued induction through 48 h (Figure 1B).

### 3.2. GLP-1R Agonists and Levonorgestrel Enhance Hormone Signaling Through the Upregulation of Steroid Hormone Receptors and Other Pathway Genes 

As shown in Figure 1, the induction of GLP-1R in response to agonist binding predicts a feed-forward loop of activity for agents such as liraglutide and semaglutide in endometrial cancer cells. Furthermore, the impact of both semaglutide and levonorgestrel on the expression of genes comprising the hormonal response pathway is substantial (Figure 2). Levonorgestrel induces GLP-1R, PGRMC1, ER, AR, MR, and PR. In the case of ER and MR, the magnitude is 40- to 95-fold. The impact on ER, AR, MR, and PR signaling is potentially enhanced given the induction of those genes. This portends the synergistic activation of the pathway when both drugs are used together.

### 3.3. Effects of GLP-1 Agonist in Combination with Levonorgestrel on Growth of Patient-Derived Organoid Models of Endometrial Cancer

To validate the potential for benefit of the combination of levonorgestrel with semaglutide in endometrial cancer treatment, we next performed studies using patient-derived models of early-stage/grade endometrial cancer. These cases are typically associated with elevated BMI and historically recognized as “Type I” endometrial cancer [27,28]. Specifically, we studied the impact of levonorgestrel, semaglutide, or the combination on cell viability in a short-term experiment in six independent patient-derived organoids (PDOs) isolated after hysterectomy from women with early-stage, grade I endometrial cancer (Table 1).

We first established expression of ER and PR in the primary tumors from which the PDO models were created (matched specimen, Figure 3). Of note, there is heterogeneity in steroid hormone receptor expression both amongst tumors and intratumorally. PR staining is primarily nuclear in all patient samples, and in some models (e.g., UBC 7104), we observed robust stromal staining of PR in addition to expression in the glandular structures.

We next assessed the impact of progestin treatment in the absence or presence of a GLP-1R agonist on PDO growth, with cell viability as the endpoint. In most cases, there was a significant reduction in cell viability by 72 h in response to exposure to 100 nM of semaglutide + 100 nM of levonorgestrel (Figure 4). Responsiveness was noted in PDOs obtained from tumors expressing relatively high PR, as well as tumors with more modest PR expression as defined by the H-score. These data confirm the activity of semaglutide in endometrial cells, as well as its potential to further reduce cell viability in the presence of levonorgestrel. The activity of this combination in some models obtained from tumors with low PR expression is of interest. Given the induction of membrane and nuclear PR expression along with other steroid hormone receptors in response to semaglutide + levonorgestrel (Figure 2), hormone responsiveness is potentially magnified even in cases that initially express low PR levels and would be considered to be hormone therapy-unresponsive using a progestin alone.

## 4. Discussion

Obesity is a risk factor for many types of cancer and a key driver in the hormonal imbalance underlying development of EC. Indeed, obesity is not only associated with increased risk of developing cancer but also with higher cancer-related mortality and death due to other comorbidities [5,6]. We now appreciate that the increase in EC incidence and mortality is driven in part by the obesity epidemic, and the healthcare burden of EC will only continue to increase if we cannot simultaneously combat obesity. Studies into the mechanisms by which obesity increases risk of EC have identified abnormal hormone levels, in particular, of estrogen, as a key driver. Specifically, androgens are converted to estrone and estradiol in adipocytes by aromatase. There is a clear link between higher aromatase levels and increased BMI in postmenopausal women [7,29]; for example, as compared to postmenopausal women with BMI in the normal range, obese age-matched women have a 40% increase in circulating levels of both estrone and estradiol [7]. In addition to excess estrogen, a reduction in circulating progesterone also occurs in obesity and can contribute to the development and/or progression of EC, such as in obese anovulatory premenopausal patients. In this case, the lack of progesterone due to anovulation mimics polycystic ovarian syndrome (PCOS), a well-known risk factor for future EC development. In addition to a decrease in the ligand, changes in expression and activity of PR are also dysregulated in EC. For example, our group has identified several epigenetic modifications (e.g., miRNAs and DNA methylation) underlying loss of PR expression in EC [30].

Based on the link between obesity and AEH/EC, we hypothesized that the combination of a GLP-1R agonist and levonorgestrel would be useful as a novel treatment or prevention regimen and tested this hypothesis in preclinical models using EC cell lines and patient-derived organoids. In these studies, we demonstrated the expression of GLP-1R in our models, which is highly relevant to the clinical situation, because GLP-1R is known to be expressed in the normal and malignant endometrium. In line with our findings, Kanda et al. reported expression of GLP-1R in Ishikawa endometrial cancer cells, as well as in over 100 tissue samples from endometrial tumors [26]. We now report that GLP-1R agonists, when combined with progestins such as levonorgestrel, have the potential to enhance progestin therapy for AEH and EC as shown in EC cell and PDO models. The treatment-related transcriptional upregulation of receptors and components of the GLP-1R/PR signaling cascade sheds light on a possible mechanism of synergy as depicted in Figure 5. Specifically, the membrane progesterone receptor component PGRMC1 has been reported to cross-talk with and enhance GLP-1-induced insulin secretion in a rat insulinoma cell line. Moreover, PGRMC1 has been shown to increase cAMP production, as well as signaling by EGFR and subsequent downstream PIK3-induced insulin secretion. The role of the membrane progesterone receptor is less defined, though it has been shown to activate PI3K-mediated survival in breast cancer, as reviewed by Camacho-Arroyo and colleagues [31]. Thus, signaling downstream of GLP-1R has the potential to create synergy between progestins and GLP-1R agonists [32]. As diagrammed in Figure 5, our data indicate that in response to semaglutide and liraglutide, the expression of GLP-1R and steroid hormone receptors, including nuclear PR, is significantly induced in endometrial cancer cell models. We hypothesize that this induced receptor expression creates a feed-forward loop of activity for agents, such as liraglutide and semaglutide, that may enhance progestin therapy. In endometrial cancer organoids, the addition of levonorgestrel to semaglutide resulted in reduced cell viability compared to either agent alone. Importantly, the impact was noted in organoids from tumors expressing both high and low levels of nuclear PR, indicating that semaglutide treatment has the potential to enhance PR activity even in tumors considered to be less sensitive to hormone therapy.

## 5. Conclusions

In summary, these studies highlight the potential usefulness of adding GLP-1R agonists to progestin therapy for AEH and EC. In addition to the obvious benefit of weight loss associated with GLP-1R agonist treatment, we show that GLP-1R agonists enhance the impact of progestin therapy, potentially through inducing hormone receptors and molecular cross-talk between GLP-1R, PGRMC1, and nuclear PRs. These findings support future clinical use of GLP-1R agonists in addition to progestins for women with AEH and EC. We propose that GLP- 1R agonists enhance progestin effectiveness, and progestins also enhance GLP-1R activation, resulting in the amplification of the effects of both agents.

## Figures and Tables

**Figure 1 cancers-17-00598-f001:**
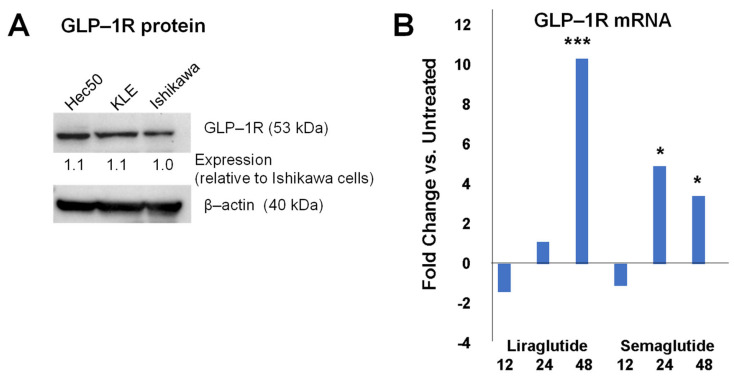
GLP-1Rs are expressed in endometrial cancer cells, and transcription is induced in response to GLP-1R agonist treatment. (**A**): Western blotting confirms protein expression of GLP-1R in multiple endometrial cancer cell lines. Expression was determined by densitometry and calculated relative to Ishikawa cells after normalization to the β-actin loading control. (**B**): qPCR for GLP receptor transcripts in Ishikawa endometrial cancer cells demonstrates expression, as well as significant upregulation of this gene in response to agonists liraglutide and semaglutide at 24 and 48 h after treatment. * *p* ≤ 0.05; *** *p* ≤ 0.001 vs. untreated control at the same timepoint.

**Figure 2 cancers-17-00598-f002:**
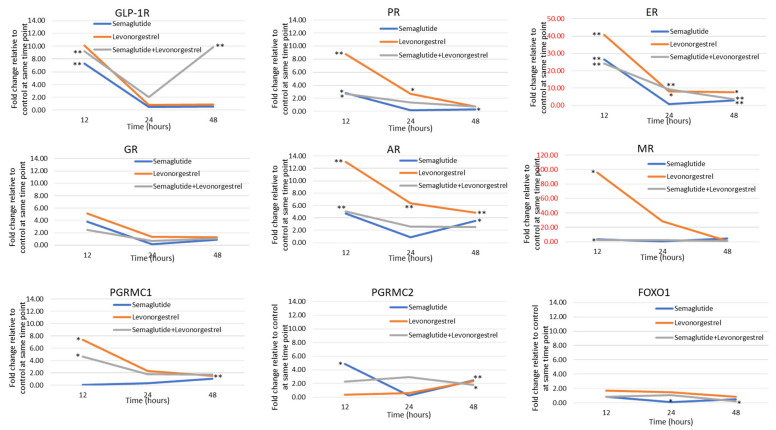
qPCR for the following mRNAs: GLP-1R, PGRMC1, PGRMC2, FOXO1, ER, AR, GR, MR, and PR following treatment of Ishikawa cells for 12, 24, and 48 h with semaglutide (blue line), levonorgestrel (orange line), or semaglutide + levonorgestrel (gray line). Fold change in expression was calculated relative to time-matched controls (DMSO vehicle) after correcting for housekeeping gene (18S rRNA) expression using the ∆∆Ct method. * *p* < 0.05; ** *p* < 0.01 vs. time-matched control by *t*-test.

**Figure 3 cancers-17-00598-f003:**
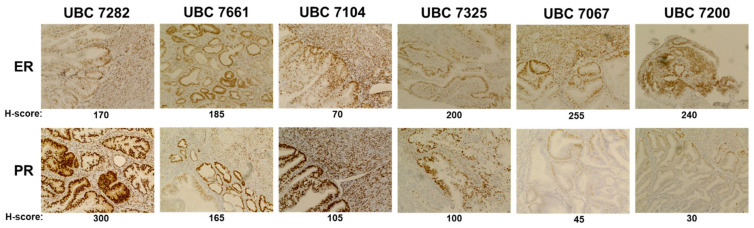
Heterogeneity in expression of estrogen receptor (ER) and progesterone receptor (PR) in early-stage/grade endometrial tumors. All images are at 20× magnification. The modified H-score for each specimen is noted below each image.

**Figure 4 cancers-17-00598-f004:**
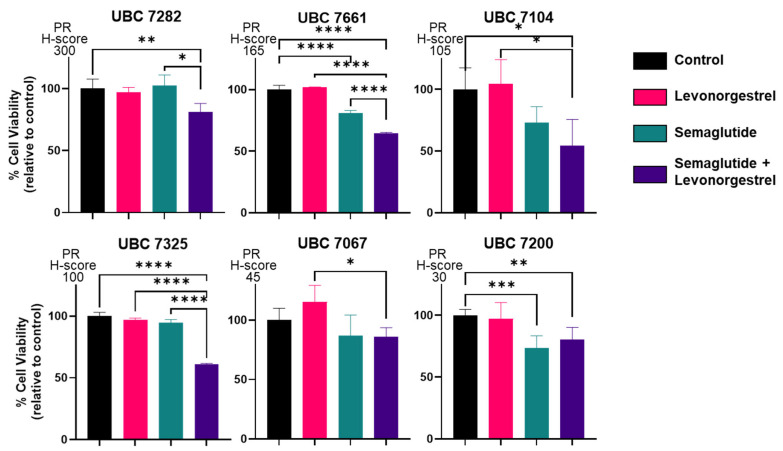
Semaglutide + levonorgestrel inhibits the growth of endometrial cancer patient-derived organoids (PDOs). PDOs from Grade 1 endometrial carcinomas were treated with progesterone (P4), levonorgestrel (L), semaglutide (S), or L+S. All drug concentrations are 100 nM. Cells were treated for 72 h. PR expression was determined by IHC of the primary tumor specimens from which the PDOs were generated (see Figure 3); models are arranged in order of highest to lowest PR expression. Control: vehicle (DMSO). * *p* < 0.05, ** *p* < 0.01, *** *p* < 0.001, or **** *p* < 0.0001 by one-way ANOVA.

**Figure 5 cancers-17-00598-f005:**
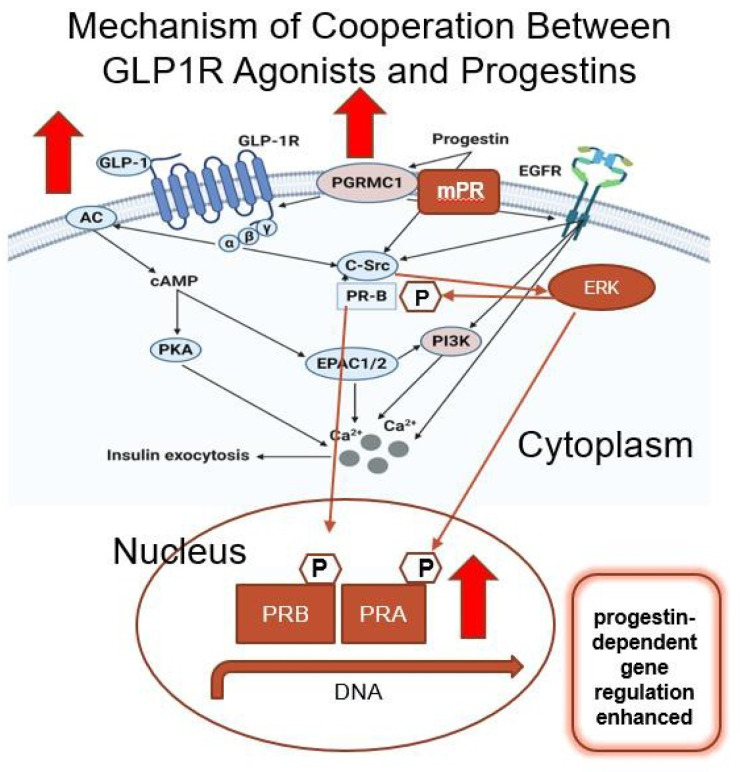
A proposed mechanism of cooperation between GLP-1R agonists and progestins. GLP-1R agonists and progestins bind to membrane receptors, including the 7-transmembrane GLP-1R and PGRMC1. It is proposed that PGRMC1 enhances the activation of GLP-1R, which signals downstream through C-Src and ERK to phosphorylate and activate PRB. That phosphorylation event induces PRB translocation to the nucleus and ligand binding, allowing activated PRB and PRA to enhance progestin-dependent gene regulation and endometrial differentiation. Activated GLP-1R also enhances cAMP effects, inducing insulin exocytosis. Red arrows denote key factors that are increased at the mRNA level by the combination of semaglutide and levonorgestrel in this study.

**Table 1 cancers-17-00598-t001:** Clinical and demographic characteristics of endometrial cancer patients from which patient-derived organoid models were created. Disease status was determined at ~3 years after primary cytoreductive surgery. NED: no evidence of disease. Estrogen receptor (ER) and progesterone receptor (PR) were assessed by immunohistochemistry (IHC) by two independent reviewers and calculated as the modified H-score as described in Materials and Methods. H-scores are the average of two independent reviewers blinded to sample identify.

ID	Age	Race/Ethnicity	Diagnosis (Endometrial Adenocarcinoma)	Primary Adjuvant Treatment	Disease Status (as of 11/2024)	ER IHC	PR IHC
UBC 7282	68	White Non-Hispanic	Stage IA/Grade 1	Clinical surveillance	Dead (metastatic cancer; cannot rule out endometrial cancer as primary site of origin)	170	300
UBC 7661	69	White Non-Hispanic	Stage IA/Grade 1	Clinical surveillance	NED	185	165
UBC 7104	66	White Non-Hispanic	Stage IA/Grade 1	Clinical surveillance	NED	70	105
UBC 7325	34	White Non-Hispanic	Stage IA/Grade 1	Clinical surveillance	NED	200	100
UBC 7067	69	White Non-Hispanic	Stage 1B/Grade 1	Vaginal brachytherapy	NED	255	45
UBC 7200	45	White Non-Hispanic	Stage IA/Grade 1	Clinical surveillance	NED	240	30

## Data Availability

All data for this manuscript are included in the primary manuscript or Appendix A.

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
