# Peer review of "Enhancing Progestin Therapy with a Glucagon-Like Peptide 1 Agonist for the Conservative Management of Endometrial Cancer"

_cancers, 2025, doi:10.3390/cancers17040598_

Round 1

Reviewer 1 Report

Comments and Suggestions for Authors

Manuscript Andrea Hagemann et al. aimed to the study the synergic effect of a GLP-1R agonist semaglutide and a progestin on endometrial cancer cell viability using cell lines and patient-derived organoids. The authors suggested that this drug combination may increase the efficacy of hormone therapy in endometrial cancer. Obtained results allowed to conclude that GLP-1R agonists synergistically enhance the impact of progestin therapy, potentially through inducing hormone receptors and through molecular cross-talk between GLP-1R, PGRMC1 and  nuclear PRs. Despite the interesting idea the authors didn’t prove to the end this conclusion.  Thus, I have many comments: 

1.      Lines 90-91: The authors claimed  that “Classic nuclear receptors are well-described transcription 90 factors, that upon phosphorylation localize in the nucleus and bind as dimers to hormone responsive gene promoters”. More correctly to say, that mainly NRs are activating by ligands (hormones etc.), some of them may also be activated upon phosphorylation. 

2.      Line 93: “membrane receptors  PGRMC1 and 2, which are activated in the cell membrane by ligand binding and”  - Reference is needed. 

3.      Ref. 26 is not completed : Thiel KW, Newtson AM, Devor EJ, Zhang Y, Malmrose PK, Bi J, Losh HA, Davies S, Smith LE, Padilla J, Leiva SM, Grueter CE,  Breheny P, Hagan CR, Pufall MA, Gertz J, Guo Y, Leslie KK. 

4.   Line 119: “in 20 µg of whole cell lysate..”. Probably authors meant µg of protein? 

5.   There is no information how protein concentration was determined and how proteins were separated. By sodium dodecyl sulphate-polyacrylamide gel electrophoresis? 

6.      Question to RT-PCR method: Data were calculated as the fold change in expression relative to untreated samples at the same time point using the ΔΔCt method. However, the authors didn’t determine the “housekeeping” gene expression (internal reference). Also there is no information how primers were designed. 

7.      Question to IHC method: no information about secondary antibodies and staining. 

8.      Incorrect citation: Kanda et al reported expression of GLP-1R in Ishikawa endometrial cancer cells 168 as well as in over 100 tissue samples from endometrial tumors (31). Ref. 31: Albitar L, Pickett G, Morgan M, Davies S, Leslie KK. Models representing type I and type II human endometrial cancers: Ishikawa 411 H and Hec50co cells. Gynecol Oncol. 2007;106(1):52- 412. Ref. 32. – is  lacking. 

9.      It is not clear why different methods for detection of receptors were used? Fig.1 Western blotting, Fig.2 – qPCR. Fig. 2 – there is no statistic. 

10.  Fig. 2,4 – not clear what was as a control. 

11.   It is not clear why three cell cultures were checking on GLP-1R, and then the Ishikawa only was used for the further research. 

12.  Ref. 7, 11 are lacking. Ref. 35 is not about PGRMC1 and GLP-1R cross-talk: Bi J, Newtson AM, Zhang Y, Devor EJ, Samuelson MI, Thiel KW, Leslie KK. Successful Patient-Derived Organoid Culture of 419 Gynecologic Cancers for Disease Modeling and Drug Sensitivity Testing. Cancers. 2021;13(12):2901. PubMed PMID: 420 doi:10.3390/cancers13122901.

13.  There is no characteristic of patient-derived EC cells. 

14.  Question to Fig.5; it is not clear why mPR taken into account? What happed with nuclear PR? Why were EGFR,PI-3K considered? 

Author Response

  1. Lines 90-91: The authors claimed  that “Classic nuclear receptors are well-described transcription 90 factors, that upon phosphorylation localize in the nucleus and bind as dimers to hormone responsive gene promoters”. More correctly to say, that mainly NRs are activating by ligands (hormones etc.), some of them may also be activated upon phosphorylation. 

Response: We have edited the text to indicated that, as this reviewer points out, nuclear receptors are activated by ligand as well as phosphorylation:

Lines 91-94: Classic nuclear receptors are well-described transcription factors that, upon ligand binding, dimerize, translocate into the nucleus and bind to hormone responsive gene promoters. In addition to ligand binding, nuclear receptors can be activated by phos-phorylation in a ligand-dependent or -independent manner.

  1. Line 93: “membrane receptors  PGRMC1 and 2, which are activated in the cell membrane by ligand binding and”  - Reference is needed. 

Response: The following reference has been added: García-Sáenz M, Ibarra-Salce R, Pozos-Varela FJ, Mena-Ureta TS, Flores-Villagómez S, Santana-Mata M, De Los Santos-Aguilar RG, Uribe-Cortés D, Ferreira-Hermosillo A. Understanding Progestins: From Basics to Clinical Applicability. J Clin Med. 2023 May 10;12(10):3388. doi: 10.3390/jcm12103388. PMID: 37240495; PMCID: PMC10218893.

  1. Ref. 26 is not completed : Thiel KW, Newtson AM, Devor EJ, Zhang Y, Malmrose PK, Bi J, Losh HA, Davies S, Smith LE, Padilla J, Leiva SM, Grueter CE,  Breheny P, Hagan CR, Pufall MA, Gertz J, Guo Y, Leslie KK. 

Response: We apologize for this and all other errors to the reference list, which occurred when the manuscript was transferred to the journal’s template. All reference errors have been corrected.

  1.  Line 119: “in 20 µg of whole cell lysate..”. Probably authors meant µg of protein? 

Response: Thank you for pointing out this error, which has been corrected. Line 121-122.

  1.  There is no information how protein concentration was determined and how proteins were separated. By sodium dodecyl sulphate-polyacrylamide gel electrophoresis? 

Response: We apologize for the brevity in methods. We now provide detailed methods for western blotting (lines 121-134):

Protein concentrations were determined using the BCA Protein Assay (Pierce/Thermo Fisher) per the manufacturer’s instructions. Proteins were separated by SDS-PAGE with 10% acrylamide followed by transfer to nitrocellulose membrane. After blocking in 5% milk in Tris-buffered saline buffer (TBST), membranes were exposed to primary anti-bodies against GLP-1R (# bs-1559R, Bioss Antibodies, 1:1000 dilution in 5% BSA in TBST, incubated overnight at 4°C) or β-actin (#A1978, Sigma, 1:5,000 dilution in 5% milk in TBST, incubated 1 hr. at room temperature), followed by exposure to either an-ti-rabbit-HRP for GLP-1R or anti-mouse HRP for β-actin (each secondary antibody was at a 1:10,000 dilution in 5% BSA in TBST for 1 hr). . Bands were visualized using HRP-conjugated chemiluminescence reagent (SuperSignal West Pico PLUS, Thermo Fisher Scientific). Expression of GLP-1R in each cell line by densitometry (NIH Image J) was calculated relative to Ishikawa cells after normalization to β-actin loading control.

  1. Question to RT-PCR method: Data were calculated as the fold change in expression relative to untreated samples at the same time point using the ΔΔCt method. However, the authors didn’t determine the “housekeeping” gene expression (internal reference). Also there is no information how primers were designed. 

Response: We apologize for this missing information. We used 18s rRNA as the housekeeping gene. We have now added a table listing all primer sequences as well as how the primers were designed (lines 141-145):

The following genes were assessed by quantitative PCR (qPCR): GLP1R, PGRMC1, PGRMC2, FOXO1, ER, AR, GR, MR, and PR, with 18S rRNA used as the housekeeping gene. The PrimerQuest primer design tool (IDT, Coralville, Iowa) was used to design primer sequences. 

  1. Question to IHC method: no information about secondary antibodies and staining.

Response: This information has been added (lines 149-162):

ER and PR expression in primary tissues from which the patient-derived organoid models were generated was assessed by IHC by the Comparative Pathology Laboratory at the University of Iowa. IHC was performed on sequential sections using the following primary antibodies from Dako: PR (PgR636); ER (M7047). Secondary antibody was goat-anti-mouse for both PR and ER. Stained sections were incubated in DAB chromogen substrate and counterstained with hematoxylin prior to mounting with coverslips. Two independent reviewers applied the modified H-score (range of 0-300O to score staining. Specifically, the modified H-score is determined by multiplying the intensity of tumor staining (0-3) by the percent of cells with staining (0-100%). Positive control tissue was scored as a 3+ for intensity and a 100% for percentage. To ensure tissue integrity, the presence of signal was assessed in the stroma and myometrium. Modified H-scores from each reviewer (N=2 reviewers) were averaged for each slide to yield the average modified H-score. Representative images were acquired with an EVOS inverted bright field microscope at 20X magnification.

  1. Incorrect citation: Kanda et al reported expression of GLP-1R in Ishikawa endometrial cancer cells 168 as well as in over 100 tissue samples from endometrial tumors (31). Ref. 31: Albitar L, Pickett G, Morgan M, Davies S, Leslie KK. Models representing type I and type II human endometrial cancers: Ishikawa 411 H and Hec50co cells. Gynecol Oncol. 2007;106(1):52- 412. Ref. 32. – is  lacking. 

Response: This has been corrected.

  1. It is not clear why different methods for detection of receptors were used? Fig.1 Western blotting, Fig.2 – qPCR. Fig. 2 – there is no statistic. 

Response: Given the number of time points and receptors assayed, we chose to use qPCR as the method to assess receptor levels. We apologize for the lack of statistical analysis in the original Figure 2, which has now been added in the revised manuscript.

  1. Fig. 2,4 – not clear what was as a control. 

Response: In all treatments, vehicle (DMSO) at an equivalent volume served as control. We have added this information to the figure legends.

  1. It is not clear why three cell cultures were checking on GLP-1R, and then the Ishikawa only was used for the further research. 

Response: Ishikawa cells are the only endometrial cancer cell line that expresses progesterone receptor (PR). Thus, to interrogate the interplay between a GLP-1 agonist and a clinically-used progestin, we focused studies on PR positive endometrial cancer cells. These studies were then followed up with studies in patient-derived organoid models with varying PR levels.

  1. Ref. 7, 11 are lacking. Ref. 35 is not about PGRMC1 and GLP-1R cross-talk: Bi J, Newtson AM, Zhang Y, Devor EJ, Samuelson MI, Thiel KW, Leslie KK. Successful Patient-Derived Organoid Culture of 419 Gynecologic Cancers for Disease Modeling and Drug Sensitivity Testing. Cancers. 2021;13(12):2901. PubMed PMID: 420 doi:10.3390/cancers13122901.

Response: This has been corrected.

  1. There is no characteristic of patient-derived EC cells. 

Response: We perform visual quality control for each organoid model to ensure they are not populated with normal endothelial cells, which will overtake the cancerous cells in culture and are easily identifiably. In general we have found that the early stage/grade endometrial lesions do not persist for extended culture. This is why we chose to present features of the patient tumor from which the organoids were created in Table 1. The amount of material available to characterize organoids will necessarily be limited to the initial passage of cells, and the cells are far less viable when one goes back to frozen stocks. Hence, multiple characterization studies are limited, which is a benefit of also studying transformed cell lines, as we report.

  1. Question to Fig.5; it is not clear why mPR taken into account? What happed with nuclear PR? Why were EGFR,PI-3K considered? 

Response: The reviewer makes very interesting points. We have now expanded the discussion to speculate about the role of mPR and EGFR/PI3K in the mechanism of synergy.

Reviewer 2 Report

Comments and Suggestions for Authors

The authors presented the article entitled “Enhancing Progestin Therapy with a Glucagon-Like Peptide 1 Agonist for the Conservative Management of Endometrial Cancer”.

Excess body weight shows varied impact on incidence of Type I and Type II endometrial cancer. I suggest Type I and/or Type II endometrial cancer addressed in the study design; If not concerned, explanation in the discussion section.

Author Response

Comment: Excess body weight shows varied impact on incidence of Type I and Type II endometrial cancer. I suggest Type I and/or Type II endometrial cancer addressed in the study design; If not concerned, explanation in the discussion section.

Response: We appreciate this suggestion to describe how our study relates to the classical distinction of endometrial cancer into type I and type II. All of the cases in this study were Specifically, we have added the following statement to the Results section, along with appropriate citations:
“…, we next performed studies using patient-derived models of early stage/grade endometrial cancer. These cases are typically associated with elevated BMI and historically recognized as “Type I” endometrial cancer.”

Reviewer 3 Report

Comments and Suggestions for Authors

In this study authors showed significant reduction in viability of EC patient derived organoids in response to combination treatment of Levonorgestrel + semaglutide.  This effect was observed in both PR high and PR low expressing tumors.  They concluded that the synergistic molecular cross talk between GLP-1R and steroid hormone receptor pathways, with potential to enhance the anti-cancer activity of levonorgestrel when combined with semaglutide.

1. References are messed up. Listed 45 but cited 35 only. Carefully verify the references and the citation order in the text.

2. Study with organoids (n=6) is interesting. The combination effect was observed in UBC7661 and UBC 7325 models only.  Is there a rational to pick only one concentration (100nM) of drugs?

3. Is cell viability study conducted with cells lines (Ishikawa, KLE and Hec50) used in the study ? It is interesting to evaluate the combination effect on the viability of the cell lines which express almost same  GLP-1R protein (Figure 1). 

4. Mechanism(s) behind the observed synergistic effect of drug combination have not been well defined.

Author Response

  1. References are messed up. Listed 45 but cited 35 only. Carefully verify the references and the citation order in the text.

Response: We apologize for this mistake, which occurred when the article was reformatted to the journal’s template. We have ensured that all references are correctly cited and the reference list aligns with the citations in the text.

  1. Study with organoids (n=6) is interesting. The combination effect was observed in UBC7661 and UBC 7325 models only.  Is there a rational to pick only one concentration (100nM) of drugs?

Response: We recognize that a dose-response curve would be ideal. However, in our experience the patient-derived organoid models are a finite resource as compared to cell lines. We perform drug testing on organoid models within 2-4 passages of initial model formation in order to avoid confounding effects due to “drift” in culture. For models for which we receive a small amount of tissue, this means that we do not have sufficient material to test a range of drug concentrations. For early stage/grade endometrial cancer in particular, the amount of tissue available for research tends to be <1 mg. This is why only one drug concentration was used for each drug in this study.

  1. Is cell viability study conducted with cells lines (Ishikawa, KLE and Hec50) used in the study ? It is interesting to evaluate the combination effect on the viability of the cell lines which express almost same  GLP-1R protein (Figure 1). 

Response: We have not performed a cell viability assay with the combination of progestin and GLP-1 agonist. This is because we have unique access to a panel of patient-derived organoids, which allow us to test the effects of the drugs as they relate to inter-patient heterogeneity than cell lines. Indeed, there is only endometrial cancer cell line that represents early stage/grade disease with intact progesterone receptor expression (Ishikawa cells).

  1. Mechanism(s) behind the observed synergistic effect of drug combination have not been well defined.

Response: Our data provide unique insight into how GLP-1 agonists alter steroid hormone receptor levels and sensitivity to progestins in endometrial cancer. Future studies will include in-depth mechanistic studies to understand the precise means by which these two pathways interact and modulate transcriptional changes. We have insured that our conclusions are not overreaching – see lines 308-317 (Discussion) and lines 329-331 (Conclusions).

Round 2

Reviewer 1 Report

Comments and Suggestions for Authors

The authors have corrected all comments. I see no more obstacles to publishing this article in “Cancers”

Reviewer 3 Report

Comments and Suggestions for Authors

Authors responded to the previous review comments and provided satisfactory explanation. There is still the references number order is not correct. All the references need to verify for the correctness of the citation with the statements in the text and update the order of references. 

After the citation of reference number 25 in page 4, the next numbers cited in the text are 29, 30 in the same page. The 26, 27, 28 references are cited after 30. These numbers need to be verified for correct reference and cite in the order.